# Observational social learning of "know-how" and "know-what" in wild orangutans: evidence from nest-building skill acquisition

Andrea L. Permana [1,2] ✉, Junaidi Jaka Permana[1], Lara Nellissen[3], Eggi Septian Prayogi[4], Didik Prasetyo[4], Serge A. Wich [5,6], Carel P. van Schaik[7,8,9] & Caroline Schuppli [2,7] ✉

Immature great apes learn how to build their nests over multiple years, yet how they do so has remained largely unclear. We investigated the detailed role of social learning in the acquisition of nest-building skills in wild Sumatran orangutans (*Pongo abelii*) using data on nest-building, nest practice, and nest peering behaviour from 44 individuals, collected over 17 years. We found that nest peering (but not being close to a nesting individual without peering) is associated with a significant increase in nest practice and is primarily directed at multi-step nest elements. Dependent immatures mostly peer at their mothers and use nest tree species in common with her, independent immatures peer at a larger range of individuals and use nest tree species in common with them. Our results suggest that orangutans acquire their nest-building skills through observational social learning, selective attention to "know-how" and the transmission of "know-what" information.

Nest-building is a critical subsistence skill in wild orangutans. Wild adult orangutans build a nest (for definition see Table 1) every evening to spend the night in and occasional day nests for rest during daylight hours[1]. Nests provide a comfortable resting place but also protect against predators, parasites, and heat loss overnight[2–4]. To produce a stable platform sufficient to support their large body weight[1,5], the construction of a nest requires relational and combinatory manipulations of materials, manual dexterity and strength[2,6]. It also requires flexible decision-making, which is likely to involve consideration of a number of trade-offs due to the multi-functional nature of nests[2].

Whereas day nests usually only consist of a frame[2], night nests often include additional comfort elements such as linings, pillows and blankets (Table 1 and Fig. 1[2,7]). Constructing these elements requires additional manipulative actions making night nests sturdier structures than day nests[2]. This is especially true for nests that are constructed using multiple trees, so-called 'multitree nests' (Table 1[7]). Twig manipulations and nest sounds are also associated with the construction of additional nest elements

(Table 1[2,7–9]). As these behaviours routinely occur in some populations but are rare or absent in others, they are classed as cultural variants[8–11].

Orangutans choose their nest tree species disproportionately to their occurrence in the habitat[2,3,12,13]. Preferred nest tree species have been shown to be especially suitable for their purpose due to the mechanical properties of their wood, leaf size, and phytochemical properties[2,14,15], which likely facilitate branch manipulations and construction security, thermal and physical comfort properties, and may or may not repel biting arthropods[2,16,17]. Selection of certain nest tree species is thus likely based on knowledge of these properties and characteristics[7,15]. The decision-making and skills required by nest-building suggest that it poses a significant cognitive challenge.

Immature orangutans begin to show interest in nest-building at around 6 months of age, in the form of play and manipulation of nesting materials, including branches, twigs, and leaves[7]. Whereas practice (Table 1 and Supplementary Video 1[18]) of day nests begins at around age 1 year and peaks at age 3–4 years, night nest practice is delayed until close to 3 years of age and

[1]Ape Tank, Department of Psychology, University of Warwick, Coventry, UK. [2]Development and Evolution of Cognition Research Group, Max Planck Institute of Animal Behaviour, Konstanz, Germany. [3]Eco-Anthropologie, Muséum National d'Histoire Naturelle, Paris, France. [4]Department of Biology, Faculty of Biology and Agriculture, Universitas Nasional, Jakarta, Indonesia. [5]School of Biological and Environmental Sciences, Liverpool John Moores University, Liverpool, UK. [6]Institute for Biodiversity and Ecosystem Dynamics, University of Amsterdam, Amsterdam, The Netherlands. [7]Department of Evolutionary Anthropology, University of Zurich, Zurich, Switzerland. [8]Department of Ecology and Animal Societies, Max Planck Institute for Animal Behaviour, Konstanz, Germany. [9]Institute for the Interdisciplinary Study of Language Evolution, University of Zurich, Zurich, Switzerland. ✉e-mail: Ani.Permana@warwick.ac.uk; cschuppli@ab.mpg.de

## Table 1 | Definitions of nesting behaviours, elements and features

| Behaviour | Definition |
| --- | --- |
| Nest | A construction consisting of branches, twigs and leaves (bent, broken, transferred) manipulated to create a resting site in a tree[78]. Functional nests were defined as nests used for resting ≥1 min after their construction. |
| Peering | Attentive close-range watching of the activities of a conspecific, sustained over at least five seconds and from a close enough distance at which the details of the peering target's activity can be seen (for the nest-building context, this is 5 m[25]). |
| Nest practice | Unsuccessful attempt at building a functional nest (by bending, breaking or intertwining leafy branches). Or seemingly successful construction of a nest without using it as a functional nest[25]. |
| Social learning | Learning that is influenced by observing, associating with, or interacting with other individuals or their products[79]. |
| Additional comfort elements | |
| Lining | Smaller branches with many leaves bent onto the nest foundation to create a layer[1], which is laid upon. |
| Pillow | Small leafy twigs arranged on one side of the nest[1], used for the head. |
| Blanket | Loose leafy branches laid on top of the body after animal lies down in the nest[1]. |
| Roof | Loose cover of braided branches woven together to make a solid, nearly waterproof, shield[1]. |
| Nest features | |
| Twig manipulation | Manipulating endings of twigs with the mouth before working them into the nest construction[1]. |
| Multitree nest | Several trees connected into a single nest by securing branches from each tree together[1]. |
| Nest sounds | Sounds produced during nest construction. At Suaq Balimbing, this includes the 'raspberry' and rarely the 'nest smack' sound[9]. |

Adapted from Permana et al.[7].

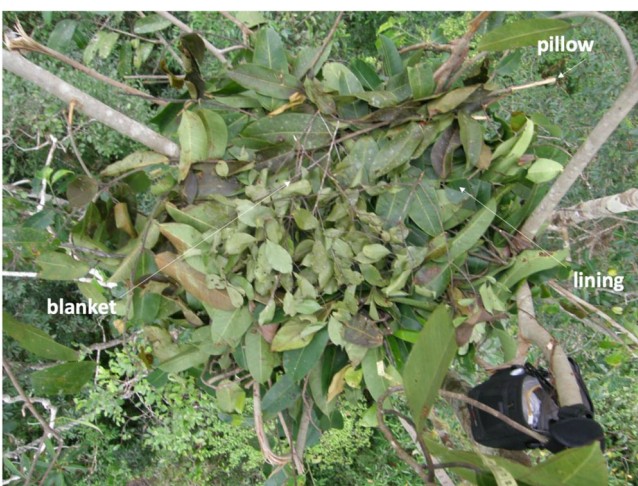

**Fig. 1 | Sumatran orangutan night nest with additional comfort elements.** A nest foundation covered with a thick lining of branches full of fresh leaves, a pillow made from Y-shaped leafy twigs, and a blanket of short leafy twigs inside a freshly abandoned arboreal nest. Photograph by Junaidi Jaka Permana.

is mastered at around 8 years[7]. Equally, the construction of additional nest elements and multitree nests occurs late in development[7]. Nest tree species repertoires and selectivity towards certain tree species increase with age, with mothers showing the largest species repertoires and selectivity for the largest number of different tree species[2,7]. These patterns suggest that young orangutans acquire the know-how needed for nest-building and information on nest tree species gradually and over multiple years.

There is evidence that social and individual forms of learning[19,20] mediate the acquisition of nest-building skills[21–25]. In terms of social learning of nest-building skills (Table 1), as with the acquisition of other subsistence skills, 'peering', which is i.e. attentively observing the behaviour of conspecifics[25–29] plays an important role. Peering at another individual building a nest (Table 1, see Supplementary Video 2[18]) is followed by increased goal-directed practice of the observed behaviour, including the investigation of nest materials and constructive behaviours or attempts thereof[7,25,30]. However, it is not yet understood which forms of social

learning are at play (e.g. whether it is peering itself or simply close proximity during peering which elicits nest practice behaviour), what kind of information is transmitted, and who individuals learn from.

Whereas previous studies provide convincing evidence of social learning in many animal species[31–33], to date, there has been a lack of differentiation between the different forms of social learning in wild study systems, particularly when it comes to teasing apart observational forms of social learning (e.g. imitation and emulation[33]) from non-observational ones (e.g. social enhancement and facilitation[34]). Observational forms of social learning likely allow for higher fidelity information transmission compared to non-observational forms and may thus facilitate the acquisition of more learning-intensive skills[34–39]. Given the higher learning intensity of multiple aspects of nest-building (see above), it is likely that nest-building is acquired via observational forms of social learning. Furthermore, so far it has not yet been investigated which elements of nest-building are acquired socially, i.e. what type of information is socially transmitted. In the foraging context, primates have been shown to socially learn how and what to eat (i.e. how to process food before ingestion and to recognise edible items[40,41]). Concordantly, in the nest-building context, individuals may peer to learn how to construct nests, including the behavioural sequences involved in the skill (i.e. to acquire so called 'know-how' information[39,42–44]) and/or what material to use to construct a nest, such as what tree species to use (i.e. to acquire so-called 'know-what' information[42,43,45,46]). In terms of who individuals learn from, research on social learning across contexts in primates suggests that over the course of development, individuals display 'biases'[31,47–49], initially learning basic skills from their most trusted role models (i.e. their primary caregivers) before gradually widening their pool of role models, which likely leads to the acquisition of additional relevant skills shown by other individuals[40]. This widening of the pool of role models can happen passively, through changing association partners, or actively through preferences to attend to specific classes of role models at certain stages of development[25,30,41,47,50,51].

In this study, we investigated the detailed role of social learning during the ontogeny of nest-building behaviour in a population of wild Sumatran orangutans (*Pongo abelii*) at the Suaq Balimbing monitoring station, building upon previous studies of the development of nest-building skills in orangutans[7,25]. The study population is known for its tendency to form temporary social groups[52,53], offering the ideal setting to study social learning, including how role model choice develops with age. Additionally,

**Table 2 | Nest peering practice cycle (Model I)**

| Factor | Factor type | Estimate | SE | *P* |
|---|---|---|---|---|
| Intercept | Intercept | −1.253 | 0.334 | <0.001 |
| Age | Control | −0.071 | 0.040 | 0.081 |
| Condition: | Predictor | | | |
| After vs. Before | | 0.987 | 0.267 | **0.001** |
| After no peer vs. Before | | 0.400 | 0.333 | 0.230 |
| After peer vs. Before | | 1.498 | 0.288 | **<0.001** |
| After peer vs. After no peer | | 1.097 | 0.304 | **0.001** |
| Individual | Random | - | - | - |

Differences in the number of immatures' nest practice events comparing the hour before the mother made a nest, the hour after the mother made a nest, the hour after the mother made a nest when the immatures had been within 5 m of the mother building her nest but were not peering, and the hour after the mother made a nest when the immatures were peering at the mother making the nest (*n* = 184 nests on 115 follow days). The age of the dependent immature was included as a control to account for changes in nest practice frequencies over age[7,25]. Analysed with a GLMM with a Poisson family distribution, including model estimates, standard errors (SE), *P*-values (*P*). The dispersion parameter was 1.156, and the ratio of observed to predicted Zeros was 0.937. Significant *P*-values of the predictors are in bold.

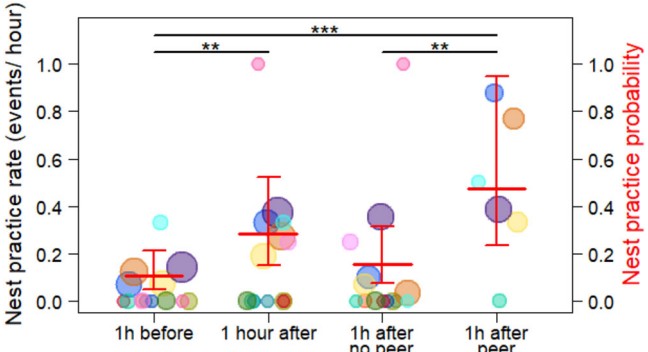

**Fig. 2 | Nest peer-practice cycle.** Dependent immatures' average nest practice rates in the hour before the mother made a nest, the hour after the mother made a nest (irrespective of the distance of the immature to its mother when she made the nest and irrespective of whether the immature was peering), the hour after the mother made a nest when the immatures had been within five metres of the mother when she made the nest but had not been peering, and the hour after the mother made a nest when the immature had been peering at the mother making the nest. The data points represent the average nest practice rate per individual and yearly age class for each condition (ranging from 0 to 1, see Methods). The size of the data points corresponds to the observed number of nest-building events of the mother for each condition and age individual class (on a log scale, *n* = 1–62 data points). The different colours represent the different focal individuals that contributed data to this analysis (*n* = 14 dependent immatures). The bold red horizontal lines represent the predicted nest practice probabilities with all other variables of the model kept constant at their mean and the thinner red lines depict the confidence intervals (based on Model I). Statistically significant differences between the conditions are indicated with * for *P*-values of less than 0.05, for less than 0.01 with ** and for less than 0.001 with ***.

the regular construction of day nests and additional comfort elements[2] provides the opportunity to investigate learning across nest types. We used peering as a behavioural measure of attendance to social information and nest practice behaviour, elicited by peering as an indicator of social learning. To investigate the attendance to 'know-how' information[39,42–44] we compared peering at different nest elements. To infer the transmission of 'know-what' information[42,43,45,46], we matched peering patterns with the overlap in nest tree species used by different individuals.

Specifically, we predicted that:

I) If nest-building skills are acquired through observational social learning, immatures will show increased rates of nest-building practice after peering at the nest-building behaviour of others (when controlling for current age-dependent skill levels). If this increase is caused by a form of observational social learning, the increase in practice rates will be higher after an immature peers at a conspecific building a nest, compared to when they are in close proximity to a conspecific building a nest but do not peer.

II) If nest peering serves to acquire information on nest-building 'know-how', controlling for age-dependent skill levels, immatures will peer most frequently at multi-step nest features as these require memorising a larger number of actions and the correct sequences thereof. We thus expect immatures to peer more at the building of night nests compared to day nests and more when nests are supported by multiple trees, contain additional comfort elements, or have used twig manipulations compared to nests not using those features.

III) If nest-building is acquired via age-dependent social learning biases, young immatures will peer mainly at their mothers whereas older immatures will gradually peer more and more at other role models[40].

IV) If 'know-what' nest-building information is socially transmitted, the overlap in the use of nest tree species among dyads will follow peering target selection. Young immatures will show a high overlap with their mother's nest tree species choices then decrease as they get older as they increasingly gather alternative information from other individuals (see prediction III). When reaching adulthood, individuals may return to the original tree species selection repertoires learnt from their mother or maintain an enhanced use of nest tree species learned from other individuals as independent immatures.

We found that peering at the nest-building events of others is associated with an increase in nest practice behaviour and is typically directed towards multi-step nest elements such as night nest-building, the construction of multitree nests, and the addition of comfort elements. Our results also show that immatures mostly peer at their mother prior to independence but then widen their pool of role models with increasing age. The use of nest tree species follows peering patterns, as dependent immatures tend to choose the same species as their mother, while independent immatures tend to choose nest tree species used by unrelated models. Together, our findings provide multiple lines of evidence for the prominent role of observational social learning, selective attention to 'know-how' and the transmission of 'know-what' information in the ontogeny of orangutan nest-building behaviour, which suggests a strong cultural element in the variation of nest-building behaviour observed across populations[54,55].

## Results

### Nest-building practice rates of immatures increase after peering

We found significant differences between rates of nest practice behaviour in dependent immatures before and after their mother made a nest, as indicated by a significant full versus null model comparison of Model I (likelihood ratio test (LRT): Chi-square = 38.049, *P* < 0.001). The full model indicated that rates of nest practice behaviour by dependent immatures significantly increased in the hour after the mother made a nest compared to the hour before. However, dependent immatures' nest practice behaviour only significantly increased in the hour after the mother made a nest when the dependent immature was peering at the mother building her nest, but not when they were within 5 m of the mother building her nest but not peering (Table 2 and Fig. 2). These effects are controlled for the age of the immature because of the known effect of age on nest practice rates[7,25].

### Immatures selectively peer at multi-step nest features

We found that the type and features of a nest had a significant effect on the probability that dependent and independent immatures peered at their mothers when she was making a nest as indicated by a significant full versus null model comparison of Model II (LRT: Chi-square = 53.033, *P* < 0.001).

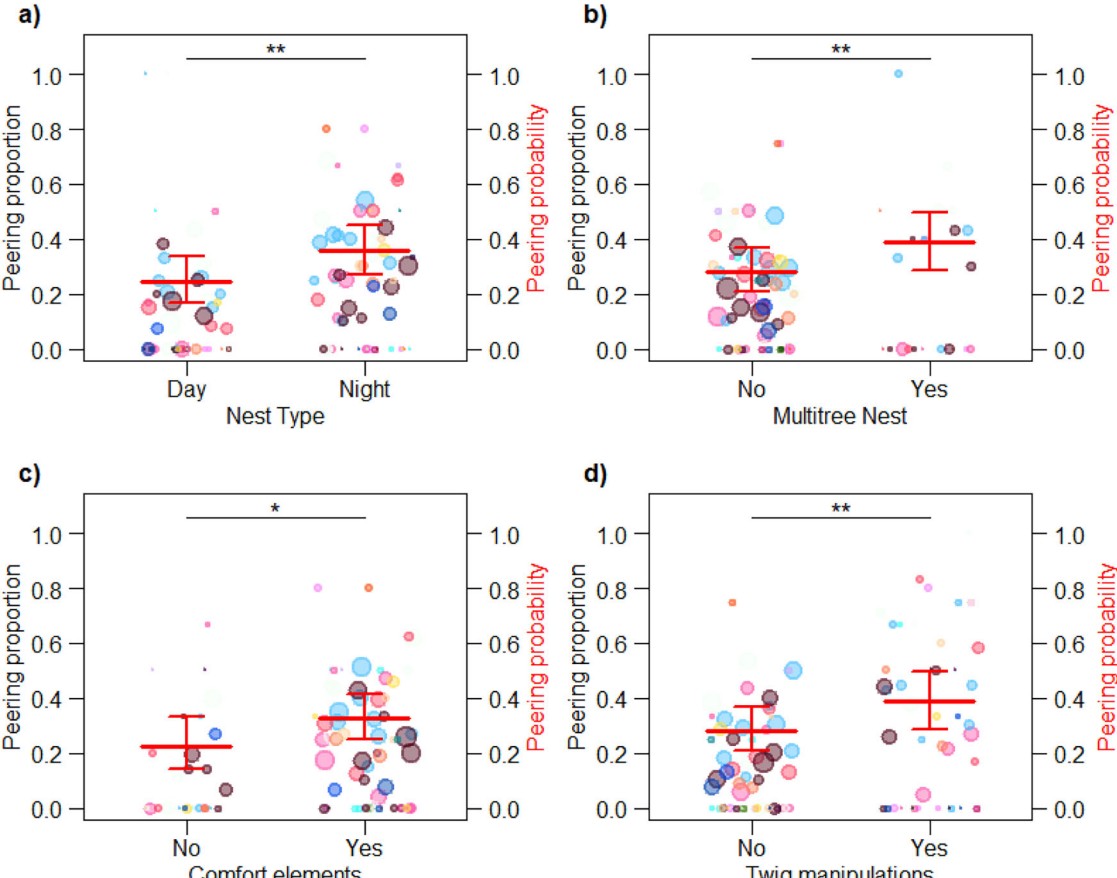

**Fig. 3 | Effects of nest features on nest peering.** Average proportion of nest-building events by the mother during which a dependent immature offspring was peering for **a** day nests versus night nests, **b** single tree versus multitree nests, **c** nests containing no comfort elements versus nests containing at least one comfort element, **d** nests for which no twig manipulations were used versus nests for which twig manipulations were used. The data points represent the average calculated peering proportions per age class and individual (see Methods). The size of the data points corresponds to the observed number of nest-building events of the mother for each condition and age individual class ($n$ = 1–79 nest-building events). The different colours represent the different focal individuals that contributed data to this analysis ($n$ = 27 immatures). The bold red horizontal lines represent the predicted peering probabilities with all other variables of the model kept constant at their mean. The thinner red lines depict the confidence intervals (based on Model II). Statistically significant differences between the conditions are indicated with * for $P$-values of less than 0.05, for less than 0.01 with ** and for less than 0.001 with ***.

The full model indicated a significantly higher probability of nest peering for night nests compared to day nests (Fig. 3a and Table 3), a significantly higher probability of nest peering for multitree compared to single tree nests (Fig. 3b and Table 3), a significantly higher probability of nest peering when the nest contained at least one comfort element compared to none (Fig. 3c and Table 3), and a significantly higher probability of nest peering when the mothers performed twig manipulations compared to when they did not—all controlled for differences in construction time between the nests (Fig. 3d and Table 3). The full model also revealed a trend for a positive effect of nest-building duration on nest peering probability, but no effect of nest sounds on nest peering probability (Table 3). Furthermore, in line with previous findings on the effect of age on nest peering[25], there was a significant effect of linear and negative quadratic age of the immature on nest peering probability, suggesting a peak followed by a decline as immatures approach independence (Table 3, see also Supplementary Fig. 1).

## Immatures widen their range of peering targets with increasing age

Overall, 83.9% of nest-peering events were directed at the mother of the peering individual. Non-mother targets included independent immatures (9.2%), unflanged males (4.1%), other adult females (1.9%), and dependent immatures (0.8%). We found that peering targets of the immatures changed with increasing age, as indicated by the nearly significant full versus null

model comparison of Model III (LRT full model versus null model: Chi-square = 3.575, $P$ = 0.059). The full model revealed a trend for a positive effect of the age of the peering immature on the probability that the peering target was an individual other than their mother (Table 4 and Fig. 4).

## Nest tree species choices of immatures follow those of the nest peering target

We found that the overlap in the use of nest tree species differed significantly between different dyads of individuals, as indicated by a significant full model versus null model comparison of Model IV (LRT: Chi-square = 51.989, $P \leq 0.001$). The full model showed that dependent immatures had a significantly higher overlap in their use of nest tree species with their mothers than independent immatures had with their mothers (Fig. 5 and Table 5). The overlap in nest tree species used by independent immatures with their mothers was significantly lower than the mothers' overlap with related adult females (Fig. 5 and Table 5). There was no significant difference in the overlap of nest tree species used by independent immatures and their mothers, compared to the overlap between independent immatures and unrelated adult females. Furthermore, related adult females (i.e. mothers) showed a significantly higher overlap in the use of nest tree species with each other, compared to unrelated adult females with each other (Fig. 5 and Table 5). Notably, the overlap in nest tree species between mothers and their dependent immatures did not differ from the overlap between related

**Table 3 | Effects of nest features on nest peering probability (Model II)**

| Factor | Factor type | Estimate | SE | *P* |
|---|---|---|---|---|
| Intercept | Intercept | −3.181 | 0.442 | <0.001 |
| Nest type (night nest) | Predictor | 0.535 | 0.204 | **0.009** |
| Multitree nest | Predictor | 0.570 | 0.213 | **0.007** |
| Comfort elements | Predictor | 0.532 | 0.235 | **0.023** |
| Twig manipulations | Predictor | 0.502 | 0.187 | **0.007** |
| Age | Control | 0.449 | 0.174 | **0.008** |
| Age² | Control | −0.045 | 0.019 | **0.018** |
| Nest sounds | Control | 0.182 | 0.186 | 0.328 |
| Nest construction duration | Control | 0.0399 | 0.022 | 0.064 |
| Individual | Random | - | - | - |
| Age individual | Random | - | - | - |
| Age² individual | Random | - | - | - |
| Individual: follow | Random | - | - | - |

The effects of nest type, multitree nests, presence of comfort elements and twig manipulations on nest peering probability, while controlling for the age of the immature individual, the presence of nest sounds, and nest-building duration. Analysed with a GLMM with a binomial family distribution (*n* = 1255 nests on 782 follow days). Including model estimates, standard errors (SE), *P*-values (*P*). Significant *P*-values of the predictors are in bold. The dispersion parameter was 0.925 and the ratio of observed to predicted zeros was 1.073.

**Table 4 | Effect of age on peering target selection (Model III)**

| Factor | Factor type | Estimate | SE | *P* |
|---|---|---|---|---|
| Intercept | Intercept | −10.389 | 1.627 | <0.001 |
| Age | Predictor | 0.454 | 0.249 | *0.069* |
| Individual | Random | - | - | - |
| Individual: follow | Random | - | - | - |

The effect of the age of the peering individual on the probability that the peering event was directed at an individual other than their own mother, analysed with a GLMM with a binomial family distribution (*n* = 371 peering events on 258 follow days). Including model estimates, standard errors (SE), *P*-values (*P*). Predictors with trend-level *P*-values are italicised. The dispersion parameter was 1.098 and the ratio of observed to predicted zeros was 0.765.

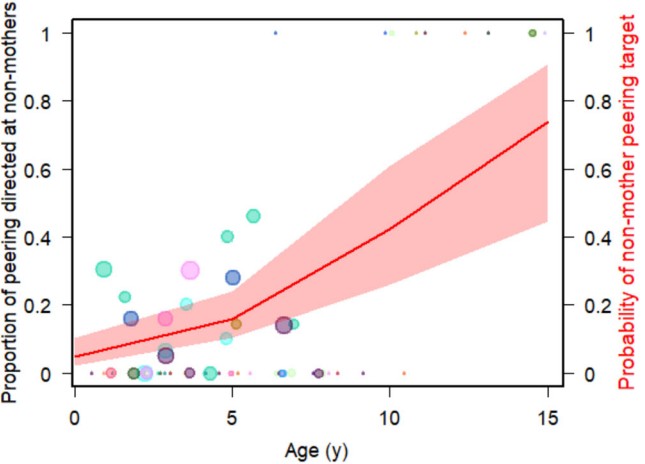

**Fig. 4 | Nest peering targets over age.** The proportion of peering directed at individuals other than the mother per individual and age class (see Methods). The size of the data points corresponds to the number of peering events based on which the proportion was calculated (on a log scale, *n* = 1–33 peering events). The different colours represent the different focal individuals who contributed data to this analysis (*n* = 25 immatures). The red line represents the predicted probability that peering is directed at a non-mother individual and the red bands represent the confidence intervals (based on Model III).

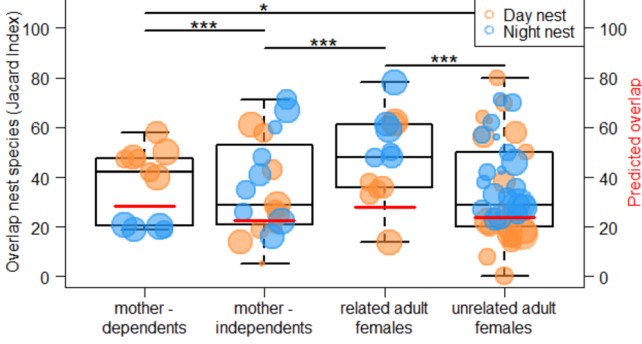

**Fig. 5 | Nest tree species overlap between individuals (Model IV).** Jaccard Similarity Index calculated for tree species overlap for mothers with their own dependent offspring, mothers with their own independent offspring, related mothers, and unrelated mothers for day nests and night nests. Each data point represents one dyad (*n* = 88 dyad combinations, including 43 unique dyads (counting each individual combination once—see Methods). The symbol size corresponds to the total average number of nests of the two individuals of each dyad (*n* = 3.5–22.5 nests). The boxes indicate the interquartile range (IQR), the central line depicts the median, the whiskers extend to 1.5 × IQR. The red horizontal lines represent the predicted overlaps with all other variables of the model kept constant at their mean (based on Model IV). Statistically significant differences between the conditions are indicated with * for *P*-values of less than 0.05, for less than 0.01 with ** and for less than 0.001 with ***.

mothers, but it was significantly higher than the overlap between unrelated mothers (Fig. 5 and Table 5). Also, across the whole dataset, the overlap in night nest tree species was significantly higher than in day nest species (Fig. 5 and Table 5). However, against this general trend, in dependent immatures, the overlap in day nest species was higher than in night nest species (Fig. 5).

## Discussion

Our results support our predictions that social learning, including observational social learning, is a prominent element in the ontogeny of orangutan nest-building behaviour, that individuals pay selective attention to nest elements that require multiple construction steps, that role model selection develops to include a wider number of individuals over age and that knowledge on the use of nest tree species is socially transmitted.

We found that peering behaviour by dependent immatures directed at their mother's nest-building events resulted in a significant increase in nest practice by dependent immatures. There was no such effect on practice for immatures who were close to their mother when she built a nest but did not peer (Table 2 and Fig. 2). Peering is an index of social learning[54] as it allows for the transfer of information between individuals. Whether information is actually transferred depends on the difference in the knowledge repertoire of the peerer and the peering target, as well as the peerer's decision or ability, to act upon the information. The significantly higher probability of nest practice after nest peering is evidence that information was transferred

during the peering event and that the observer then acted upon the information they observed. In line with the effect of nest peering on nest practice, daily nest peering rates followed a similar age trajectory as nest practice rates (with a peak around the age of 4–5 years), and by the time individuals were reliably building fully functional night nests (see Table 1), nest peering and nest practice had stopped[7,25] (Supplementary Fig. 1). Notably, the immatures' probability to be within peering distance while their mother was building a nest decreased continuously between the ages of 3 and 9 years, following the general development of the distance between immatures and their mothers during daily activities over age[21,24,56]. As such, nest peering distance during nest building events and nest peering probabilities showed

**Table 5 | Nest tree species overlap between different dyads (Model IV)**

| Factor | Factor type | Estimate | SE | *P* |
|---|---|---|---|---|
| Intercept | Intercept | −0.877 | 0.113 | <0.001 |
| Dyad type | Predictor | | | |
| Mother–dep. vs. Mother–indep. | | −0.244 | 0.045 | **<0.001** |
| Mother–dep. vs. Mother–related | | −0.146 | 0.059 | 0.804 |
| Mother–dep. Vs. Mother–unrel. | | −0.370 | 0.143 | **0.038** |
| Mother–indep. Vs. Mother–related | | 0.230 | 0.048 | **<0.001** |
| Mother–indep. Vs. Mother–unrel. | | −0.126 | 0.140 | 0.739 |
| Mother–rel. vs. Mother–unrel. | | −0.355 | 0.142 | **0.038** |
| Nest type (night) | Predictor | 0.049 | 0.022 | **0.024** |
| Nr. shared + unsh. sp. | Offset | - | - | - |
| Dyad | Random | - | - | - |

Differences in nest tree species use between different classes of dyads (mothers and their own dependent immatures (dep.) versus mothers and their own independent immatures (indep.), mothers and their own dependent immatures versus related mothers (rel.), mothers and their own dependent immatures versus unrelated mothers (rel.), mothers and their own independent immatures versus related mothers, mothers and their own independent immatures versus unrelated mothers, and mothers (unrel.) versus unrelated mothers). versus night nest type on nest tree species overlap analysed with a GLMM with a Poisson family distribution (*n* = 88 dyad combinations, including 43 unique dyads counting each individual combination once—see Methods). The number of shared nest tree species by each dyad was used as the response and the sum of shared and unshared species as an offset. Shown are estimates, standard errors (SE), and *P*-values. Significant *P*-values of the predictors are in bold. The dispersion parameter was 0.725 and the ratio of observed to predicted zeros was 0.496.

distinct age trajectories, illustrating that nest peering is more than a passive result of being close to their mothers (Supplementary Fig. 2).

Since no increase in nest practice was observed after the immatures had been close to their mother when she was building a nest but did not peer, non-observational forms of social learning are not supported. However, we are currently unable to determine which specific form/s of observational social learning (e.g. imitation, emulation, or observational conditioning[34,57,58]) are operating, mainly due to the observational nature of our data[59]. Specific experimental work may be able to identify which of these forms of observational social immature orangutans use to acquire their nest-building skills.

Generally, it is difficult to separate learning 'know-how' from 'know-what' information through observational studies in natural contexts because knowing how to perform a skill in most cases also requires knowing what to perform the skill with[35]. However, the variation we see in orangutan nest-building in terms of the use of additional nest elements and nest tree species use offers a unique opportunity for us to attempt to distinguish between types of socially transmitted information.

If immatures acquire information on how to construct their nests, we expect them to pay most attention to multi-step elements of nest-building as these require memorising a larger number of individual steps and sequences thereof. We found that multi-step features of nest-building, including the construction of night nests, the use of multitree nests, the construction of additional comfort elements, and twig manipulations elicited the highest peering rates (controlling for nest-building duration, i.e. peering opportunities, Table 3 and Fig. 3), evidencing selective attention to 'know-how' information[39,42–44]. However, our study cannot provide direct evidence for the actual transmission of 'know-how' information. This may become possible by analysing the details of how individuals execute the different steps to construct multi-step nest features, given that there is sufficient individual variation. Nevertheless, these results provide indirect evidence that observational social learning is particularly important for the

acquisition of the more intricate elements of nest-building and that these features require more extensive observation in order to be learned. Peering rates were lower for simpler nesting behaviours such as the construction of day nests, single tree nests, and night nests without comfort elements and twig manipulations (Table 3 and Fig. 3). These results are consistent with what is known about the acquisition of common and complex skills required for food processing and tool use[21,25,56,60].

We found that immatures peered almost exclusively at their mothers, but with increasing age, nest peering was directed more towards non-mother role models (Table 4 and Fig. 4). We know that based on the mothers' stable association patterns, from around 2 years on, immatures have equal opportunities to peer at their mothers' association partners throughout the rest of their dependency period[47]. Furthermore, at that age, immatures' locomotor skills have increased[24], allowing them to approach association partners to peer at independently of their mothers. Yet despite these equal opportunities, we see an increase in peering at role models other than the mother with age throughout the dependency period (Fig. 4), suggesting an increasing preference for non-mother role models. After the age of 8–10 years, older immatures roam free of their mothers[21], associate more with other individuals, and therefore have more opportunities to peer at others[61]. During this time, nest peering is almost exclusively directed at individuals other than the mother. Our data set does not allow us to control for peering opportunities during the independent immature phase thus, we cannot say whether the patterns we have observed are based on independent immature preference or if they are due to shifts in role model availability. However, analyses of peering in the feeding context show that increasing peering at non-mother role models persists throughout the entire immature period when controlling for the time immatures spend in association with each type of role models[25,47].

Our results show that orangutans use vertical and oblique social learning[34,62] to acquire their nest-building skills, supporting the three-phases of social learning framework for primates for a variety of learning contexts proposed by Whiten and van de Waal[40]. During the first phase of nest-building ontogeny, as with the development of diet and foraging preferences[29,56], it is safest to follow the most trusted and readily available role model, namely the mother (Table 4 and Fig. 4). As immature orangutans get older and begin to increase their independence from their mother, other role models become more important as they may possess skills and knowledge that are still new to the immature[40,47]. In the final phase of the Whiten and van de Waal framework[40], social learning continues into independence and into adulthood. To investigate social learning in the nest-building context during adulthood, one could look at peering by migrating unflanged males upon their arrival in a new area, as has been done in the feeding context[51].

We found that dyadic overlaps in individuals' use of nest tree species between age classes change with age (Fig. 5 and Table 5). Dependent immatures have a significantly higher overlap with their mother in their use of nest tree species than independent immatures. Independent immatures have a higher overlap with unrelated adult females than with their mothers. As such, the nest tree species overlap follows peering patterns, which suggests that the choices for particular nest tree species, i.e. 'know-what' information[39,42–44] is learned socially. However, to disentangle whether the transmission of 'know-what' information requires actual peering as opposed to merely being in association (i.e. observational versus non-observational forms of social learning), a more detailed data set would be needed.

An increasing proclivity for non-mother role models with age suggests that while immatures may acquire information about suitable nest tree species from observations of their mother, learning from additional non-mother role models and likely also independent practice, improves and expands knowledge of suitable nest trees over time. Adaptively, this makes up for incomplete repertoire transmission from mothers to their offspring and upskills individuals to use a wider range of tree species to independently explore if these species are suitable nest trees and potentially discover additional benefits, such as beneficial phytochemical properties of some species[2,16,17].

Tree species overlap among individual mothers varied according to relatedness (Fig. 5 and Table 5). Related mothers showed a significantly higher overlap in nest tree species choice than unrelated ones (Fig. 5 and Table 5). Strikingly, the overlap between related mothers is significantly higher than between mothers and their independent offspring (Fig. 5 and Table 5). This means that after going through a phase of intensive practice and learning from others during the independent immature period, individuals at some point in adulthood revert back to their mothers' choices. This indicates that the development of nest-building skills is not complete until adulthood[7] and that 'know-what' information learned during early immaturity shapes adult choices[39,43,44].

Across orangutan populations, related and unrelated females show highly overlapping home ranges although related females spend more time in association with one another than they do with other females[63,64]. At Suaq, females live at the highest densities and form regular social parties with close maternal relatives, which provides them with ample opportunities for horizontal and oblique transmission throughout life[65,66]. The higher overlap in nest tree species choices among related adult females thus follows association patterns and suggests that nest tree species choices are socially learned and transmitted across generations, i.e. they are likely cultural. The local variation seen in several features of nest-building across orangutan populations[54,55] is also in line with this interpretation. Ecological effects, such as gradients in habitat availability of the different tree species on nest tree species choices remain to be investigated. However, the high overlap of the home ranges of unrelated adult females at Suaq makes it unlikely that these effects cause the patterns observed here. Even though we cannot rule out the possibility of a genetic explanation, such as through biases in manipulative propensities, the genetic mechanism this would require (i.e. genes coding for propensities to manipulate certain tree species) does not seem plausible[40].

Overall, the overlap in nest tree species choices between dyads was significantly higher for night nests than for day nests (Table 5). This concurs with a smaller number of tree species that are used for night nests than for day nests (identifiable day nest tree species, n = 32, Supplementary Table 2; night nest trees, n = 30 species, Supplementary Table 3). Night nests are sturdier constructs (see Introduction) and need to be more durable and comfortable (because they are used for a longer duration) as well as keep the individual warm during cooler nighttime weather. Our results support the idea that fewer tree species are suitable for building nests that live up to these requirements. Even though the size of our dataset did not allow for statistically testing to see if the pattern of a higher overlap for night nest compared to day nest species holds across all comparisons, Fig. 5 suggests that for the nest tree species overlap between dependent immatures and their mothers, the pattern went in the opposite direction. Throughout dependency, immatures maintain close spatial proximity to their mothers[21,24]. Night nest construction by dependent immatures often occurs during or after the mother has constructed her night nest. At these times, dependent immatures are likely to remain particularly close to the mother because they eventually join her in her night nest. It is therefore likely that a closer physical proximity in the evening results in more similar nest tree species choices for night nests.

In conclusion, our study sheds novel light onto the social learning processes underlying the acquisition of nest-building skills in wild orangutans. Our results showed that: (1) Nest peering, but not close proximity to a nesting individual alone, leads to a selective increase in nest practice, which supports observational forms of social learning but not non-observational ones. (2) Complex elements of nest-building (i.e. night nests, multitree nests, additional comfort elements and twig manipulations) elicit more peering than simple elements (i.e. day nests, single tree nests), suggesting that individuals pay selective attention to 'know-how' information. (3) Immatures peer increasingly towards non-mother role models as they get older, suggesting that individuals optimise skill acquisition to learn the most relevant information at each developmental stage[40]. (4) Overlaps in the use of nest tree species follow the age-specific peering patterns in that the overlap in nest tree species between mothers and their offspring decreases during the independence period. Furthermore, related adult females use more similar nest tree species than unrelated adult females, suggesting that nest tree species choices (i.e. 'know-what' information) are mainly learned via social learning from the mother during dependency, followed by social learning from a wider range of role models and individual exploration during juvenility. Across generations, this may lead to cultural variation in nest-building elements and thus the differences in aspects of nest-building observed across populations[54,55,67]. The results of this study are further evidence that social and individual learning pervades the immature phase of orangutan's lives[25,29,47,51,68–70].

## Materials and methods
### Data collection
Data were collected by 59 experienced observers on wild Sumatran orangutans at Suaq Balimbing (3° 02.873′ N, 97° 25.013′E) between 2007 and 2024. The 550 ha study area is a peat swamp forest, with mixed dipterocarp and riverine forest located in the South Kluet region of the Gunung Leuser National Park in Nanggroe Aceh Darussalam (NAD), Sumatra, Indonesia[52]. New observers at the station undergo data collection training, and their data is only included once a concordance index of more than 85% has been reached with experienced observers during simultaneous follows.

Our full data set included 44 recognised individuals: 13 mothers (one of which was followed from independence and one from dependence) and 27 immatures. Of these, 19 were followed only as dependent immatures and 6 individuals followed only as independent immatures. Our data also included six individuals followed from dependence through to independence, one individual followed as an independent and as a mother and one individual followed from dependence throughout independence to motherhood. Details on the focal individuals included in our analyses are summarised in Supplementary Table 1. Immature animals were classed as dependent (constantly travelling with their mother: around 0–8 years) or independent (observed at least once without their mother for a minimum of 3 consecutive days but not yet at reproductive age: around 8–15.5 years). The average age at independence of the individuals in our data set was 8.6 years (6.6–8.9). Previous estimates state an average weaning age of 7–9 years at Suaq and age at first reproduction as 15–16 years[71].

Standardised activity data recorded at 2-min intervals were collected by trained researchers and field assistants during focal animal follows via instantaneous sampling (www.ab.mpg.de/571325/standarddatacollectionrules_suaq_detailed_jan204.pdf). Focal animals were followed opportunistically upon encountering them in the forest and for a maximum of 10 consecutive days, after which another focal animal was sought. Consequently, our data often have gaps of several months where individuals were not seen. In addition to activity data, we collected all-occurrence data on nest-building, peering events and nest practice behaviour (see Table 1 for definitions). For each such event, details including the nest-building duration, nest type (day nest or night nest), and nest tree species were recorded on standardised data sheets. This included whether the nest was made in a single or using multiple trees and whether the nest-builder made a lining, pillow, blanket, or roof (collectively referred to as additional comfort elements, Table 1 and Fig. 1), whether twigs of the nest were manipulated with the mouth before incorporating them (twig manipulations) and if nest sounds were made during the construction process. All-occurrence data on nest practice behaviour were collected by a subset of the observers and therefore, sample sizes vary between our different analyses.

Ethical approval for our research was granted by the Indonesian Institute of Science (LIPI), the Indonesian State Ministry for Research and Technology (RISTEKDIKTI), the National Research and Innovation Agency (BRIN), the Directorate General of Natural Resources and Ecosystem Conservation under the Ministry of Environment and Forestry of Indonesia (KSDAE-KLHK) and the Gunung Leuser National Park (TNGL).

### Nest tree species use
For each nest, we identified the species of the supporting tree. Multitree nests, or nests constructed using more than one tree for the support,

accounted for 16% ($n = 3564$) of all nests in the sample, regardless of species. In all of these cases it was possible to accurately identify which of the trees represented the 'main' supporting tree, that is which tree offered the principal supporting branch to the frame of the nest. In the analysis of nest tree species, we therefore only included the principal nest tree species as an indicator of tree species choice. Tree species identities were confirmed using samples collected in the field with the National Herbarium of Indonesia. Owing to a lack of phenology and samples of botanical species in riverine and hill forest zones at the borders of the study area, nests constructed in these areas and those made in unknown species were removed from tree species choice analysis ($n = 376$ nests, including $n = 29$ nests in *Neesia sp.* and $n = 147$ unknown nest trees). This gave us a sample of 818-day nests (Supplementary Table 2) and 892-night nests (Supplementary Table 3; Total $n = 1710$).

## Statistics and reproducibility

All analyses and plots were done using R version 4.2.2[72]. Data were analysed using Generalised Linear Mixed Models (GLMM's) as implemented in the lme4 package[73]. To test if immatures showed increased rates of nest-building practice after peering at nest-building behaviour (Prediction I, Model I), we analysed whether the number of nest practice events by dependent immatures increased in the hour after their mother started building a nest compared to the hour before when the immature was either (a) in proximity (i.e. within 5 m) of its mothers but not peering or (b) in proximity and peering. Note that we restricted this analysis to nests of the mothers of the focal immatures to avoid confounding effects of differences in levels of tolerance by individuals other than the mother. We used a GLMM with a Poisson family distribution and the number of nest practice events before and after the mother's nest-building events on a follow day as a response variable and the number of nests on that day as an offset.

To analyse if immatures peer more frequently at the building of night nests compared to day nests and more frequently at nests that contain learning-intensive features compared to nests that contain no such features (Prediction II, Model II), we analysed the effects of nest properties on the immatures' peering probability. For this analysis, we used nest-building events of mothers when they were associated with their dependent or independent immatures. We did not include nest-building events of individuals other than the mother to avoid potential confounding effects of differences in levels of tolerance. We used a GLMM with a binomial family distribution and whether the immature was peering or not (dummy coded as 0 = no peering, 1 = peering) as a response variable.

To investigate whether older immatures increasingly peer at role models other than their mother (Prediction III, Model III), we analysed the effect of the age of the immatures on peering target selection. We used a GLMM with binomial family distribution and peering target class (dummy coded as 0 = mother and 1 = non-mother individual) as a response variable.

For Models I and II, we included the age of the immature as a control to account for changes in nest practice/peering frequency over age which are likely caused by advancing skill levels[7,25]. For Model II, we also included nest-building duration and the presence of nest sounds as controls (i.e. fixed effects that were not directly connected to our predictions) because longer nest-building duration allows for more peering, and nest sounds may make nest-building more salient.

In analyses of Models I–III, several focal individuals contributed to multiple observations. To avoid pseudo-replication and account for systematic inter-individual differences, the focal individual was included as a random intercept in Models I–III. In Model II, we also included the age of the focal individual as a random slope to account for potential individual age trajectories. Model convergence issues, most likely caused by the complexity of the models in relation to the sample size, prevented us from including the age of the focal individuals as a random slope in Models I and III. In Models II and III, we included follow (i.e. follow number) as a random intercept nested within focal individual, to account for the fact that in these analyses, there were multiple data points per follow (each follow is on one focal

individual). Details of Models I–III can be seen in the Results (Tables 2–4). Individuals that contributed data points to the different analyses are listed in Supplementary Table 1.

To visualise the data, we used nest practice rates (i.e. counts per hour) for Model I, and peering proportions (i.e. proportion of events with versus without peering) for Models II and III. To be able to calculate these rates/proportions, for each individual, we grouped all available data into age-yearly classes by rounding an individual's age at each data point to the closest full yearly age. In the plots, we used the weighted average age of the individual (according to the available data points during the age window (see Results).

To investigate whether nest tree species choices follow those of the peering targets (Prediction IV, Model IV), we calculated similarities in nest tree species choices between individual dyads via the Jaccard Similarity Index (JSI; Supplementary Fig. 3). The Index ranges from 0-100%; with a higher percentage indicating higher similarity in the nest tree species used between individuals and dyads. To test for statistical differences in nest tree species use between different types of dyads of individuals (according to their age class), we used a GLMM with a Poisson family distribution. We used the number of shared species of each dyad as a response variable and the sum of the number of shared and unshared species as an offset. Because the number of species used by individuals differed between dyads ($n$ per individual = 3–20 species), we included the average number of species of the two individuals of each dyad as a weight in the analysis. Due to the longitudinal character of data, the same dyads of individuals could occur as different types of dyads (i.e. for some individuals, we had data on nest tree species use when they were dependent immatures, independent immatures, and mothers and therefore some individuals occurred in the dependent immature-mother, independent immature-mother and related mother's dyads). To avoid pseudo-replication and account for systematic differences between dyads, we included the dyad as a random effect in this analysis. The details and variables of Model IV are listed in Table 5 and the individuals that contributed data points to the data set are listed in Supplementary Table 1. The results of this analysis were visualised by plotting the Jaccard Similarity Index (Fig. 5).

For all GLMM models, to assess the overall effect of our predictors, we tested the full model (including all predictors, controls, and random effects) against the null model (including the controls and random effects only) with a likelihood ratio test via the anova function[74,75]. Due to the known non-linear effect of age on nest peering and nest practice behaviour[7,25], in Models I and II, using a model comparison via LRT, we tested whether including age as a quadratic and linear effect increased the model fit compared to including it as a linear effect only. We then proceeded with the supported model as the final model. In a last step, we looked at the significance of the individual predictors as directly retrieved from the model output of the final models. We investigated differences between the multi-level categorical variables (i.e. 'condition' in Model I and 'dyad type' in Model IV) using post hoc tests as implemented in the glht function of the multcomp package[76]. For all GLMMs, the DHARMa package in R[77] was used to test for over- / under-dispersion and zero inflation using the testDispersion and testZeroInflation functions. We found no evidence for dispersion issues or zero inflation in any of the models.

## Reporting summary

Further information on research design is available in the Nature Portfolio Reporting Summary linked to this article.

## Data availability

All data needed to reproduce the results presented in this paper are available on zenodo[18].

## Code availability

All analyses in this work were carried out using R software version 4.2.2[72] and the following packages: lme4[73], multcomp[76], DHARMa[77]. The R code to reproduce the results presented in this article is available on zenodo[18].

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

## Acknowledgements

We thank all students, researchers and field assistants who contributed to the collection of the nest data. We thank the authorities in Indonesia for their support of our work and their permission to conduct our research, including: the Indonesian State Ministry for Research and Technology (RISTEKDIKTI), the National Research and Innovation Agency (BRIN), the Directorate General of Natural Resources and Ecosystem Conservation—Ministry of Environment & Forestry of Indonesia (KSDAE-KLHK), the Ministry of Internal affairs, and Gunung Leuser National Park (TNGL). We also thank the local team, in particular Pak Arif Saifudin, S. Si, Ibu Fitriana, Samsul Kamal, and Pak Zakir. We are grateful to the Yayasan Ekosistem Lestari (YEL) and the Sumatran Orangutan Conservation Program (SOCP) for hosting our project at the Suaq Balimbing monitoring station. We are also grateful to the Faculty of Biology at the National University (UNAS) in Jakarta for their collaboration and valuable support for our research, in particular Dr. Sri Suci Utami Atmoko and Dr. Tatang Mitra Setia. This research was funded by the University of Zurich, Max Planck Institute of Animal Behaviour, AH Schultz Stiftung, PanEco Foundation, Mensch und Tier Stiftung F.i.Br., and the Leakey Foundation (Primate research fund). C.S. was supported by a Freigeist fellowship of the Volskswagen Foundation.

## Author contributions

A.P., C.P.v.S. and C.S. conceived and designed the study. A.P., C.P.v.S. designed the study. A.P., J.J.P., L.N., E.S.P. and C.S. conducted the investigation, collected and helped to curate the data. A.P. and C.S. analysed and visualised the data. S.A.W. and C.P.v.S. supervised the research. A.P. and C.S. wrote the manuscript. A.P., C.S., J.J.P., D.P., C.P.v.S. and S.A.W. reviewed and edited the manuscript.

## Competing interests

The authors declare no competing interests.

## Ethics

We the authors affirm that this research was conducted in accordance with the highest ethical standards in biological research. All applicable institutional and national guidelines for the study of wild animals were followed. Ethical approval for our research was granted by the Indonesian Institute of Science (LIPI), the Indonesian State Ministry for Research and Technology (RISTEKDIKTI), the National Research and Innovation Agency (BRIN), the Directorate General of Natural Resources and Ecosystem Conservation under the Ministry of Environment and Forestry of Indonesia (KSDAE-KLHK) and the Gunung Leuser National Park (TNGL). All authors are committed to fostering an inclusive, collaborative research environment. Contributions to this work were made regardless of gender, race, ethnicity, nationality, disability, sexual orientation, or background. All authors support equity, diversity, and inclusion in science and acknowledge the importance of promoting underrepresented voices in biological research. We also confirm that the data and materials supporting this publication are made openly available in accordance with journal policies, promoting transparency and reproducibility.
