## [Transparent Peer Review file · Communications Biology]

Observational social learning of “know-how” and “know-what” in wild orangutans: Evidence from nest building skill acquisition

Corresponding Author: Dr Andrea Permana

Version 0:

Reviewer comments:

Reviewer #1

(Remarks to the Author)

This paper describes the role of “peering” in the development of nest-building behavior in orangutans, and argues that this behavior underlies the acquisition of knowledge both of how to build the nest (“know-how”) and of what trees to use to build it (“know-what”). Moreover, the paper suggests that while dependent orangutans primarily rely on learning from their mothers, independent individuals privilege learning from other individuals. Overall, this paper provides compelling evidence for the extensive observational learning involved in the development of this sophisticated and skilled behavior, highlighting its potential for cultural variability, and I think its findings will be of interest to those studying the development and acquisition of complex behaviors in other animals.

I have no substantial critiques of the content of the paper, as I think the authors successfully leveraged 17 years of data on nest-building behavior to obtain a large enough sample size to support their claims. Below are my suggestions for improving it, most of them stylistic:

Abstract:

- Add a comma in the first sentence after “multiple years”
- Delete “and” and add a comma in “is associated with a significant increase in nest practice behavior, and mostly directed at multi-step nest elements”

Introduction:

- Line 31: add a comma after “frame”
- Line 32: I would include the definitions of the various components of the nest in the main body of the text rather than in a separate table in the supplemental materials, as it adds to readers’ understanding of the complexity of nest-building behavior.
- Lines 51-52: Rephrase to “The decision-making and skill required by nest building suggest that it poses a significant cognitive challenge.”
- Line 64: Use either “nest building” or “nest-building” consistently
- Line 65: Change to “which is i.e. attentively observing”
- Lines 98-99: Change to “and nest practice behavior, elicited by following peering as an indicator of social learning.”
- Line 100: Add “of” after “transmission”

Results:

- Increase image resolution of the graphs (also the one in the Supplemental materials).
- Line 348: Reference to supplemental figure is incomplete, it just says “Figure”
- Line 352: Close the parenthesis after “conditioning”
- Line 434: Add a comma after “populations”

Supplementary Materials:

- Table 1: Change semi-colon to period after the definition of “Peering”.

Reviewer #3

(Remarks to the Author)

Review of the manuscript titled "Observational social learning of "know-how" and "know-what" in wild 2 orangutans: evidence from nest building skill acquisition" for publication.

The manuscript is good for showing new aspects of developing nest-building behavior in wild orangutans. The present article focuses on "peering," one of the hot topics of current studies on social learning and cultural propagation in wild orangutans and chimpanzees. The authors have published three related articles.

1, Reference #6 in this manuscript

Permana, A. L. Cognitive and Cultural Aspects of Nest Building Behaviour in Wild Orangutans. (University of Amsterdam, 2022). It seems to be important, but I can not access it.

2, Reference # 24

Schuppli, C., Meulman, E. J., Forss, S. I., Aprilinayati, F., Van Noordwijk, M. A., & Van Schaik, C. P. (2016). Observational social learning and socially induced practice of routine skills in immature wild orang-utans. *Animal Behaviour*, 119, 87-98.

According to the highlights, the article revealed

Wild immature orangutans learn their routine skills by observing others (peering).

Most peering happens in the feeding and nest-building contexts.

Peering is followed by selective practice of the observed behavior.

With increasing age, the preference for role models other than the mother increases.

Thus, the critical points of the current manuscript were somehow already mentioned in this article. The current manuscript is an important addition to the details of the peering behavior.

3, Reference #7

Permana, A. L., Permana, J. J., Nellissen, L., Prasetyo, D., Wich, S. A., van Schaik, C. P., & Schuppli, C. (2024). The ontogeny of nest-building behaviour in Sumatran orang-utans, *Pongo abelii*. *Animal Behaviour*, 211, 53-67.

Highlights

Orang-utans gradually acquire nest-building skills over around 7 years.

Day nests are practiced and mastered earlier and quicker than night nests.

Multistep features of nest building appear late in development.

Orang-utans show greater tree species selectivity for night than day nests.

The development of nest tree species selection suggests these are learnt socially.

* According to the follow-up published erratum, "the last highlight, currently 'Development of nest tree species selection suggests these are learnt socially', should read 'Development of nest tree species selection suggests preferences are learnt'. We should have changed this highlight during revision when we removed the social learning element of the paper, but it was overlooked."

This is the sister article of the current manuscript, written by the same first author. Although there are a few overlaps, the two articles complement each other. I believe the current manuscript can be a good contribution to understanding the nesting behavior of orangutans.

The current manuscript says

In conclusion, our study sheds novel light on the social learning processes underlying the acquisition of nest-building skills in wild orangutans.

1. Nest peering, but not close proximity to a nesting individual alone leads to a selective increase in nest practice which supports observational forms of social learning but not non-observational ones.
2. Complex elements of nest building (i.e., night nests, multitree nests, additional comfort elements and twig manipulations) elicit more peering-practice cycles than simple elements (i.e., day nests, single tree nests), suggesting that individuals pay selective attention to "know-how" information.
3. Immatures peer increasingly towards non-mother role models as they get older, suggesting that individuals optimize skill acquisition to learn the most relevant information at each developmental stage
4. Overlaps in the use of nest tree species follow the age-specific peering patterns in that the overlap in nest tree species between mothers and their offspring decreases during the independence period.

In summary, I like this manuscript, and it is worth publishing. However, there are three significant major concerns about this study and the related study by this group of authors.

1, The data collection.

The original data was collected by 59 people between 2007 to 2024. Lines 508-510, it says, "Standardized activity data recorded at 2-minute intervals were collected by trained researchers and field assistants during focal animal follows via instantaneous sampling:

(www.ab.mpg.de/571325/standarddatacollectionrules_suaq_detailed_jan204.pdf).

The data collection protocol is very detailed. However, does this kind of data set sound enough? Have the authors tried to test the "reliability" between different observers? For example, there should be a comparison of your data and the data collected by the other local assistant(s). Longitudinal observation is very important and precious, but there might be an effort to check its reliability.

For example, in the case of chimpanzees, there are 27 years of the dataset of nut-cracking behavior (Berdugo et al., 2024, *Nature Human Behavior*): see Berdugo, S., Cohen, E., Davis, A. J., Matsuzawa, T., & Carvalho, S. (2024). Reliable long-term individual variation in wild chimpanzee technological efficiency. *Nature Human Behaviour*, 1-9. However, the dataset of this study is the video archives. Therefore, you can replicate the observation by analyzing the same videotapes repeatedly. In my experience, the orangutans in the wild stay very high in the trees so that video recording might be difficult. However, the authors show a good example of video recording (2024, March 1st). I hope you make an additional effort. The partial data can be based on the more scientifically sound data.

2, Peering and Distance

Suppose that an infant orangutan was with her/his mother. The distance between the mother and infants is a function of age and context. See Mendonça et al. (2017, *Primates*) (Mendonça, R. S., Kanamori, T., Kuze, N., Hayashi, M., Bernard, H., & Matsuzawa, T. (2017). Development and behavior of wild infant-juvenile East Bornean orangutans (*Pongo pygmaeus morio*) in Danum Valley. *Primates*, 58, 211-224.), It reported developmental changes in mother-offspring distance, contact, and activity budgets in Bornean orangutans aged 1 to 7 years. Mother-offspring contact lasted longer in resting contexts; contact during traveling was almost non-existent after 4 years of age. Comparisons with previously published data on the Sumatran species *Pongo abelli* revealed no fundamental differences in these behavioral measures. There is a qualitative difference between the two periods of development: 0 to 3 years old in close contact and 4 to 7 years old in no contact. The closeness to the mother should define this kind of developmental stage. Therefore, I understand peering is almost inevitable 100% at 0 to 3 years old, and true “peering” is getting the matter at four and later.

3, Distinction of “know how” and “know what.”

Is this a helpful dichotomy? What the current manuscript found is not necessarily related to this kind of dichotomy. It was not clear why the authors kept these terms to explain the results. I recommend that the authors change the title of the manuscript and describe what they found.

The Zone of Latent Solution (ZLS) is a topic in cumulative culture. The manuscripts cited two articles (#38 and #39) for know-how and know-what. To discuss the details, please see the following articles and the controversy:

*Tennie, C., Call, J., & Tomasello, M. (2009). Ratcheting up the ratchet: on the evolution of cumulative culture. *Philosophical Transactions of the Royal Society B: Biological Sciences*, 364(1528), 2405-2415.

**Tennie, C., Bandini, E., Van Schaik, C. P., & Hopper, L. M. (2020). The zone of latent solutions and its relevance to understanding ape cultures. *Biology & Philosophy*, 35, 1-42.

***Koops et al (2022, *Nature Human Behaviour*) (see Koops, K., Soumah, A. G., van Leeuwen, K. L., Camara, H. D., & Matsuzawa, T. (2022). Field experiments find no evidence that chimpanzee nut cracking can be independently innovated. *Nature Human Behaviour*, 6(4), 487-494.).

****Tennie, C., & Call, J. (2023). Unmotivated subjects cannot provide interpretable data and tasks with sensitive learning periods require appropriately aged subjects: A Commentary on Koops et al.(2022)“Field experiments find no evidence that chimpanzee nut cracking can be independently innovated”. *Animal Behavior and Cognition*, 10(1), 89-94.

The terms “know-what” and “know-how” were introduced by Tennie and his colleagues (see Tennie and Call, 2023, for example). Under the ZLS hypothesis, you can do the experimental study of captive orangutans (or chimpanzees) by manipulating the experimental factors. However, it may lack the ecological validity of testing orangutan behavior. In contrast, like the present manuscript, field observation has rich information on the behavior in their natural habitat. However, it is difficult to answer questions such as “know-what” and “know-how.” In my understanding, one cannot meaningfully separate know-what from know-how for an embedded resource in an ecologically valid setting (Koops et al., 2022).

Let us imagine the real life of orangutans in the wild. The mother-infant pair may encounter the old nests in the forest. What do they do? The present article focuses on the nest-building behavior of orangutans and the observational learning by the young. However, there might be another opportunity to learn. Please imagine the situation of only nests there and no behavior of nest-building: Old nests. Focusing on the “play” behavior of the old nests may shed new light on the development of nesting behavior.

4, Minor points to be corrected

Line 50, “may repel biting arthropods”

See Koops, K., McGrew, W. C., de Vries, H., & Matsuzawa, T. (2012). Nest-building by chimpanzees (*Pan troglodytes verus*) at Seringbara, Nimba Mountains: antipredation, thermoregulation, and antivector hypotheses. *International Journal of Primatology*, 33, 356-380.

The manuscript measured mosquito densities at ground level and in trees at 10 m and related mosquito densities to nesting patterns. It found no support for mosquito densities to explain arboreal nest-building. The thermoregulation hypothesis was supported. Therefore, the present manuscript should say “ may or may NOT repel biting arthropods,” too.

Line 750, miss typing

Reference 49.

Forss, S. & van Schaik, Prof. Dr. C. Social learning and independent exploration in immature Sumatran Orangutans. *Anthropological Institute vol. MSc (University of Zurich, 2009).*

*The author's name and title are strange.

** misspelling of “independent”

I hope these comments are helpful for the revision.

Version 1:

Reviewer comments:

Reviewer #1

(Remarks to the Author)

I thought this article was very strong upon first review, and I believe this revised version is now ready for publication as the authors have incorporated all of my suggested edits. The only one that wasn't incorporated may be due to a miscommunication because of how I phrased it. I suggested that the sentence in the abstract should be "is associated with a significant increase in nest practice behaviour, mostly directed at multi-step nest elements." That is, without the "and". I also think in the title the first word after the colon should be capitalized ("Evidence", not "evidence"), though that may depend on the journal's specific style.

Reviewer #3

(Remarks to the Author)

The authors investigated the role of social learning in the acquisition of nest-building skills in wild Sumatran orangutans (*Pongo abelii*). The study is based on data from 44 individuals collected over 17 years (2007 to 2024). 59 experienced observers were involved in the data collection. The authors focused on the importance of "peering." According to their definition, it is "Attentive close-range watching of the activities of a conspecific, sustained over at least five seconds and from a close enough distance at which the details of the peering target's activity can be seen (for the nest-building context, this is 5 meters". Peering was compared with non-peering, which is being close to a nesting individual without peering. As a result, peering was associated with a significant increase in nest practice behavior and was mainly directed at multi-step nest elements. Dependent immatures mostly peer at their mothers and use nest tree species in common with them; independent immatures peer at a more extensive range of individuals (non-mothers) and use nest tree species in common with them. I understand that the nest-building behavior can be a good example of discriminating between "know-how how to make)" and "know-what (what kind of trees)." I also understand the authors wanted to keep the subtitles about social learning, including selective attention to "know-how" and transmitting "know-what" information.

Decision on manuscript COMMSBIO-24-7361

Dear Reviewers,

Thank you for taking the time to review our manuscript, “Observational social learning of “know-how” and “know-what” in wild orangutans: evidence from nest-building skill acquisition”.

Your constructive feedback and insightful comments have been invaluable in helping us improve the quality and clarity of our work.

We are grateful for your thorough evaluation and the thoughtful suggestions provided. Your expertise has helped us refine and strengthen our analysis, and we have taken great care in addressing each of your comments in our revised manuscript.

Thank you once again for your support in the peer-review process. Your contributions are essential to maintaining the integrity and rigor of scholarly research, and we truly value your input.

Yours sincerely,

The Authors.

Reviewers' comments:

Reviewer #1 (Remarks to the Author):

This paper describes the role of “peering” in the development of nest-building behavior in orangutans, and argues that this behavior underlies the acquisition of knowledge both of how to build the nest (“know-how”) and of what trees to use to build it (“know-what”). Moreover, the paper suggests that while dependent orangutans primarily rely on learning from their mothers, independent individuals privilege learning from other individuals. Overall, this paper provides compelling evidence for the extensive observational learning involved in the development of this sophisticated and skilled behavior, highlighting its potential for cultural variability, and I think its findings will be of interest to those studying the development and acquisition of complex behaviors in other animals.

I have no substantial critiques of the content of the paper, as I think the authors successfully leveraged 17 years of data on nest-building behavior to obtain a large enough sample size to support their claims. Below are my suggestions for improving it, most of them stylistic:

- We thank the reviewer for the positive assessment and the constructive comments!
-

Abstract:

- Add a comma in the first sentence after “multiple years”

- Actioned

- Delete “and” and add a comma in “is associated with a significant increase in nest practice behavior, and mostly directed at multi-step nest elements”

- Actioned

Introduction:

- Line 31: add a comma after “frame”

- Actioned

- Line 32: I would include the definitions of the various components of the nest in the main body of the text rather than in a separate table in the supplemental materials, as it adds to readers’ understanding of the complexity of nest-building behavior.

- Thank you for this valuable comment. After careful consideration we have decided that as the Table of definitions is rather large, a good compromise was to move the Table to the main body of the manuscript from the Supplementary materials instead of including them in the main body of text.
- We would like to query the formatting of the reference to where this table has been adapted from. Previously, we used the number reference style only, but this seems unclear and so have added the name of the reference with the number as well. Please advise on the style to follow here.

- Lines 51-52: Rephrase to “The decision-making and skill required by nest building suggest that it poses a significant cognitive challenge.”

- Actioned

- Line 64: Use either “nest building” or “nest-building” consistently

- Thank you for recognising this inconsistently. We have revised this to the compound noun of “nest-building” across the manuscript.

- Line 65: Change to “which is i.e. attentively observing”

- Actioned

- Lines 98-99: Change to “and nest practice behavior , elicited by following peering as an indicator of social learning.”

- Actioned

- Line 100: Add “of” after “transmission”

- Actioned

Results:

- Increase image resolution of the graphs (also the one in the Supplemental materials).
 - We are sorry that the quality of the graphs was poor. We believe that this must be an artifact of the automatically generated submission pdf as the resolution of the originals is high. We will upload the graphs separately in this submission round, hoping that this will take care of the issue.

- Line 348: Reference to supplemental figure is incomplete, it just says “Figure”
 - We have fixed this error to refer to Supplementary Figure 1 and have included two references which had also disappeared since an earlier version.

- Line 352: Close the parenthesis after “conditioning”
 - Actioned

- Line 434: Add a comma after “populations”
 - Actioned

Supplementary Materials:

- Table 1: Change semi-colon to period after the definition of “Peering”.
 - Actioned

Reviewer #2 (Remarks to the Author):

Review of the manuscript titled “Observational social learning of “know-1 how” and “know-what” in wild

2 orangutans: evidence from nest building skill acquisition” for publication.

The manuscript is good for showing new aspects of developing nest-building behavior in wild orangutans. The present article focuses on “peering,” one of the hot topics of current studies on social learning and cultural propagation in wild orangutans and chimpanzees. The authors have published three related articles.

1, Reference #6 in this manuscript

Permana, A. L. Cognitive and Cultural Aspects of Nest Building Behaviour in Wild Orangutans. (University of Amsterdam, 2022). It seems to be important, but I can not access it.

- Please try this link https://pure.uva.nl/ws/files/88655787/Front_matter.pdf.
- Now number 1.

2, Reference # 24

Schuppli, C., Meulman, E. J., Forss, S. I., Aprilinayati, F., Van Noordwijk, M. A., & Van Schaik, C. P. (2016). Observational social learning and socially induced practice of routine skills in immature wild orang-utans. *Animal Behaviour*, 119, 87-98.

According to the highlights, the article revealed

Wild immature orangutans learn their routine skills by observing others (peering).

Most peering happens in the feeding and nest-building contexts.

Peering is followed by selective practice of the observed behavior.

With increasing age, the preference for role models other than the mother increases.

Thus, the critical points of the current manuscript were somehow already mentioned in this article. The current manuscript is an important addition to the details of the peering behavior.

3, Reference #7

Permana, A. L., Permana, J. J., Nellissen, L., Prasetyo, D., Wich, S. A., van Schaik, C. P., & Schuppli, C. (2024). The ontogeny of nest-building behaviour in Sumatran orang-utans, *Pongo abelii*. *Animal Behaviour*, 211, 53-67.

Highlights

Orang-utans gradually acquire nest-building skills over around 7 years.

Day nests are practiced and mastered earlier and quicker than night nests.

Multistep features of nest building appear late in development.

Orang-utans show greater tree species selectivity for night than day nests.

The development of nest tree species selection suggests these are learnt socially.

* According to the follow-up published erratum, “the last highlight, currently ‘Development of nest tree species selection suggests these are learnt socially’, should read ‘Development of nest tree species selection suggests preferences are learnt’. We should have changed this highlight during revision when we removed the social learning element of the paper, but it was overlooked.”

This is the sister article of the current manuscript, written by the same first author.

Although there are a few overlaps, the two articles complement each other. I believe the current manuscript can be a good contribution to understanding the nesting behavior of orangutans.

The current manuscript says

In conclusion, our study sheds novel light on the social learning processes underlying the acquisition of nest-building skills in wild orangutans.

1. Nest peering, but not close proximity to a nesting individual alone leads to a selective increase in nest practice which supports observational forms of social learning but not non-observational ones.

2. Complex elements of nest building (i.e., night nests, multitree nests, additional comfort elements and twig manipulations) elicit more peering-practice cycles than simple elements (i.e., day nests, single tree nests), suggesting that individuals pay selective attention to “know-how” information.

3. Immatures peer increasingly towards non-mother role models as they get older, suggesting that individuals optimize skill acquisition to learn the most relevant information

at each developmental stage

4. Overlaps in the use of nest tree species follow the age-specific peering patterns in that the overlap in nest tree species between mothers and their offspring decreases during the independence period.

In summary, I like this manuscript, and it is worth publishing. However, there are three significant major concerns about this study and the related study by this group of authors.

- We thank the reviewer for the positive assessment! As the reviewer points out, building up on the abovementioned papers, the current manuscript makes the following critical novel contributions:
 - 1) We differentiate between the different forms of social learning by analysing the effects of peering versus close proximity without peering on nest practice. Through this we find compelling evidence for learning through observation under natural contexts.
 - 2) We pin down which elements of the nest-building context elicit social attention which leads to novel insight into the type of information that is transmitted during nest peering, namely that orangutans likely peer to learn how to construct nests.
 - 3) We show that in the nest-building context, peering at non-mother role models increases with age. So far, this result was found for orangutan peering in the feeding context but not peering in the nest-building context.
 - 4) We successfully link data on the development of nest tree species selection with data peering role model selection, leading to evidence for the transmission of information on nest tree species selection through nest peering.

1, The data collection.

The original data was collected by 59 people between 2007 to 2024. Lines 508-510, it says, "Standardized activity data recorded at 2-minute intervals were collected by trained researchers and field assistants during focal animal follows via instantaneous sampling: (www.ab.mpg.de/571325/standarddatacollectionrules_suaq_detailed_jan204.pdf).

The data collection protocol is very detailed. However, does this kind of data set sound enough? Have the authors tried to test the "reliability" between different observers? For example, there should be a comparison of your data and the data collected by the other local assistant(s). Longitudinal observation is very important and precious, but there might be an effort to check its reliability.

For example, in the case of chimpanzees, there are 27 years of the dataset of nut-cracking behavior (Berdugo et al., 2024, *Nature Human Behavior*): see Berdugo, S., Cohen, E., Davis, A. J., Matsuzawa, T., & Carvalho, S. (2024). Reliable long-term individual variation in wild chimpanzee technological efficiency. *Nature Human Behaviour*, 1-9. However, the dataset of this study is the video archives. Therefore, you can replicate the observation by analyzing the same videotapes repeatedly. In my experience, the orangutans in the wild stay very high in the trees so that video recording might be difficult. However, the authors show a good example of video recording (2024, March 1st). I hope you make an additional effort. The partial data can be based on the more scientifically sound data.

- We thank the reviewer for their thoughtful comment. We now elaborate in the Methods section, (L515-519) that “New observers at the station undergo multiday data collection training by following focal individuals together with experienced observers. The data of new observers is only included once a concordance index of more than 85% has been reached with experienced observers during simultaneous interobserver follows during which there is no exchange between the new and the experienced observer.” This allows us to monitor and assess the reliability of each observer’s data taken in the field, with comparisons across local and international researchers and assistants.
- We fully agree with the reviewer that video data would make our data reproducible. We are impressed by the video archives on chimpanzee behaviour which continue to reveal compelling findings. As noted by the reviewer, it is very difficult to take good video recordings of orangutans in the wild, particularly during nest-building events because most nest-building events take place high up in dense trees (on average at or above 15 meters) and often when it is getting dark. Our video archive on nest-building events is growing but hasn’t yet reached a sufficient size for analyses. We appreciate the comment and are continually trying to improve our methods in the field, however we are constrained by the conditions of the habitat.

2, Peering and Distance

Suppose that an infant orangutan was with her/his mother. The distance between the mother and infants is a function of age and context. See Mendonça et al. (2017, Primates) (Mendonça, R. S., Kanamori, T., Kuze, N., Hayashi, M., Bernard, H., & Matsuzawa, T. (2017). Development and behavior of wild infant-juvenile East Bornean orangutans (*Pongo pygmaeus morio*) in Danum Valley. *Primates*, 58, 211-224.), It reported developmental changes in mother-offspring distance, contact, and activity budgets in Bornean orangutans aged 1 to 7 years. Mother-offspring contact lasted longer in resting contexts; contact during traveling was almost non-existent after 4 years of age. Comparisons with previously published data on the Sumatran species *Pongo abelli* revealed no fundamental differences in these behavioral measures. There is a qualitative difference between the two periods of development: 0 to 3 years old in close contact and 4 to 7 years old in no contact. The closeness to the mother should define this kind of developmental stage. Therefore, I understand peering is almost inevitable 100% at 0 to 3 years old, and true “peering” is getting the matter at four and later.

- This is an interesting point – thank you for raising this.
- We included an additional analysis to explore the relationship between the age trajectories of the immatures’ probabilities to peer at their mothers’ nests and the immatures’ probabilities to be within peering distance of their mother’s nests. We found that the probability to peer at the mother’s nest increases in early dependency, peaks at the age of 4.5 years and then decreases. The probability to be within peering distance during the mother’s nest-building events is very high in early development and then gradually drops to very low values between the ages of 3 and 9 years. This shows that peering is not simply a result of being within peering distance. However, we agree that being close to the mother, especially during early development certainly facilitates peering. We included the plots and models of these additional results in the Supplementary Information and pick them up in the discussion section (lines 354-363 of the new version of the manuscript, Supplementary Figure 2, and Supplementary Table 4). In this new section, we now also cite the work by Mendonça et al.

3, Distinction of “know how” and “know what.”

Is this a helpful dichotomy? What the current manuscript found is not necessarily related to this kind of dichotomy. It was not clear why the authors kept these terms to explain the results. I recommend that the authors change the title of the manuscript and describe what they found.

The Zone of Latent Solution (ZLS) is a topic in cumulative culture. The manuscripts cited two articles (#38 and #39) for know-how and know-what. To discuss the details, please see the following articles and the controversy:

*Tennie, C., Call, J., & Tomasello, M. (2009). Ratcheting up the ratchet: on the evolution of cumulative culture. *Philosophical Transactions of the Royal Society B: Biological Sciences*, 364(1528), 2405-2415.

**Tennie, C., Bandini, E., Van Schaik, C. P., & Hopper, L. M. (2020). The zone of latent solutions and its relevance to understanding ape cultures. *Biology & Philosophy*, 35, 1-42.

***Koops et al (2022, *Nature Human Behaviour*) (see Koops, K., Soumah, A. G., van Leeuwen, K. L., Camara, H. D., & Matsuzawa, T. (2022). Field experiments find no evidence that chimpanzee nut cracking can be independently innovated. *Nature Human Behaviour*, 6(4), 487-494.).

****Tennie, C., & Call, J. (2023). Unmotivated subjects cannot provide interpretable data and tasks with sensitive learning periods require appropriately aged subjects: A Commentary on Koops et al.(2022)“Field experiments find no evidence that chimpanzee nut cracking can be independently innovated”. *Animal Behavior and Cognition*, 10(1), 89-94.

The terms “know-what” and “know-how” were introduced by Tennie and his colleagues (see Tennie and Call, 2023, for example). Under the ZLS hypothesis, you can do the experimental study of captive orangutans (or chimpanzees) by manipulating the experimental factors. However, it may lack the ecological validity of testing orangutan behavior. In contrast, like the present manuscript, field observation has rich information on the behavior in their natural habitat. However, it is difficult to answer questions such as “know-what” and “know-how.” In my understanding, one cannot meaningfully separate know-what from know-how for an embedded resource in an ecologically valid setting (Koops et al., 2022).

Let us imagine the real life of orangutans in the wild. The mother-infant pair may encounter the old nests in the forest. What do they do? The present article focuses on the nest-building behavior of orangutans and the observational learning by the young. However, there might be another opportunity to learn. Please imagine the situation of only nests there and no behavior of nest-building: Old nests. Focusing on the “play” behavior of the old nests may shed new light on the development of nesting behavior.

- We see the reviewer’s critique on the concept of dividing social learning according to the type of transmitted information, including Know-How versus Know-What, etc. information. The broad categorization of socially transmitted information according information types has been used in a similar way in the context of diet learning in previous works including work by Rapaport and Brown (1) and our own work (2).

- Notably, we use this concept independent of the ZLS hypothesis – it was only connected to the ZLS hypothesis at a later point in time. We have updated the citations in the text where we introduce the concept to make this clearer. Even-though we largely agree with reviewer’s critique on the concept, we do believe that in its original form, it is useful when studying social learning in wild animals. We believe that the concept is particularly useful to study ape nest-building where individuals need to learn distinct types of information including how to construct a nest (e.g., the behavioural sequences involved in nest building), what materials they can use for nest-building (e.g., the tree species) and where they can build their nests (e.g., in which branches and positions within a tree/s). We decided to keep the title of our manuscript as it is but edited the text sections in which we introduce the concept on the different types of socially transmitted information in the introduction (see lines 88-94) to increase clarity on how we use the concept. We now also discuss the limitations of the concept as pointed out by the reviewer in the discussion section (see lines 373-378) and cite Koops et al. 2022.
- We agree that old or relic nests remaining in the landscape may offer learning opportunities for young orangutans. This has yet to be formally investigated. Indeed, this would support the view that nests function as tools. However, inspection of old nests is also part of wild orangutans’ foraging behaviour as they search for ants and other such arthropods, which quickly colonise abandoned nests. Because the incidence of old nest inspection is extremely low, we cannot analyse these events, but it is definitively something we should do in the future once the data set on nest inspection has grown to a larger size.

4, Minor points to be corrected

Line 50, “may repel biting arthropods”

See Koops, K., McGrew, W. C., de Vries, H., & Matsuzawa, T. (2012). Nest-building by chimpanzees (*Pan troglodytes verus*) at Seringbara, Nimba Mountains: antipredation, thermoregulation, and antivector hypotheses. *International Journal of Primatology*, 33, 356-380.

The manuscript measured mosquito densities at ground level and in trees at 10 m and related mosquito densities to nesting patterns. It found no support for mosquito densities to explain arboreal nest-building. The thermoregulation hypothesis was supported. Therefore, the present manuscript should say “ may or may NOT repel biting arthropods,” too.

- Actioned.

Line 750, miss typing

Reference 49.

Forss, S. & van Schaik, Prof. Dr. C. Social learning and independent exploration in immature Sumatran

Orangutans. *Anthropological Institute vol. MSc (University of Zurich, 2009).*

*The author's name and title are strange.

** misspelling of “independent”

- Actioned.

I hope these comments are helpful for the revision.

- We thank both reviewers again for their time in considering our manuscript and for their very constructive feedback.

Yours sincerely,
The Authors.

AUTHOR RESPONSE TO REVIEWERS' COMMENTS:

Dear reviewers,

We would like to thank reviewers for their constructive and thoughtful reviews of our manuscript entitled, "Observational social learning of "know-how" and "know-what" in wild orangutans: Evidence from nest-building skill acquisition." We appreciate the time and expertise invested in evaluating our work. We have carefully addressed all comments and suggestions and believe the manuscript is significantly improved as a result. Below we provide a point-by-point response to each reviewer's feedback, outlining the changes we have made and the rationale behind them.

We are grateful for the opportunity to revise and resubmit and we hope the revised manuscript meets your expectations.

Thank you again for your consideration.

Sincerely,

The Authors.

Reviewer #1 (Remarks to the Author):

I thought this article was very strong upon first review, and I believe this revised version is now ready for publication as the authors have incorporated all of my suggested edits. The only one that wasn't incorporated may be due to a miscommunication because of how I phrased it. I suggested that the sentence in the abstract should be "is associated with a significant increase in nest practice behaviour, mostly directed at multi-step nest elements." That is, without the "and". I also think in the title the first word after the colon should be capitalized ("Evidence", not "evidence"), though that may depend on the journal's specific style.

- We have not removed the 'and' because this alters the point that we are trying to make. We want to say here that I) peering is associated with an increase in nest practice **and** II) peering is mostly directed at multi-step nest elements.
- We have therefore slightly rephrased the sentence to clarify this. "We found that nest peering (but not being close to a nesting individual without peering) is associated with a significant increase in nest practice and is primarily directed at multi-step nest elements." We hope this is now clearer.

Reviewer #3 (Remarks to the Author):

The authors investigated the role of social learning in the acquisition of nest-building skills in

wild Sumatran orangutans (*Pongo abelii*). The study is based on data from 44 individuals collected over 17 years (2007 to 2024). 59 experienced observers were involved in the data collection. The authors focused on the importance of "peering." According to their definition, it is "Attentive close-range watching of the activities of a conspecific, sustained over at least five seconds and from a close enough distance at which the details of the peering target's activity can be seen (for the nest-building context, this is 5 meters". Peering was compared with non-peering, which is being close to a nesting individual without peering. As a result, peering was associated with a significant increase in nest practice behavior and was mainly directed at multi-step nest elements. Dependent immatures mostly peer at their mothers and use nest tree species in common with them; independent immatures peer at a more extensive range of individuals (non-mothers) and use nest tree species in common with them. I understand that the nest-building behavior can be a good example of discriminating between "know-how how to make)" and "know-what (what kind of trees)." I also understand the authors wanted to keep the subtitles about social learning, including selective attention to "know-how" and transmitting "know-what" information.

- We thank the reviewer for their helpful comments which have significantly improved our manuscript, and we are pleased the reviewer finds our decision to retain our title favourable.